# Liquid Reservoir Weld Defect Detection Based on Improved YOLOv8s

**DOI:** 10.3390/s25216521

**Published:** 2025-10-23

**Authors:** Zonghang Li, Tao Song, Bin Zhou, Yupei Zhang, Shifan Yu, Songxiao Cao, Zhipeng Xu, Qing Jiang

**Affiliations:** 1College of Metrology Measurement and Instrument, China Jiliang University, Hangzhou 310018, China; s23020804039@cjlu.edu.cn (Z.L.); zhoubin@cjlu.edu.cn (B.Z.); caosongxiao@cjlu.edu.cn (S.C.); xuzhipeng@cjlu.edu.cn (Z.X.); 06a0203051@cjlu.edu.cn (Q.J.); 2Zhejiang Institute of Quality Science, Hangzhou 310018, China; beibei210363@126.com (Y.Z.); shyfan2002@163.com (S.Y.)

**Keywords:** liquid reservoir welds, defect detection, YOLOv8s

## Abstract

The liquid reservoir is a critical component of the automotive air conditioning system, while weld seams on its surface may exhibit different types of defects with various shapes and scales, meaning traditional detection methods struggle to detect them effectively. In this article, we propose a YOLOv8s-based algorithm to detect liquid reservoir weld defects. In order to improve feature fusion within the neck and enhance the model’s capacity to detect defects showing substantial size variations, the neck is optimized through the integration of the improved Reparameterized Generalized Feature Pyramid Network (RepGFPN) and the addition of a small-object detection head. To further improve the capacity of identifying complex defects, the Spatial Pyramid Pooling Fast (SPPF) module in YOLOv8s is substituted with Focal Modulation Networks (FocalNets). Additionally, the Cascaded Group Attention (CGA) mechanism is incorporated into the improved neck to minimize the propagation of redundant feature information. Experimental results indicate that the improved YOLOv8s achieves a 6.3% improvement in mAP@0.5 and a 4.3% improvement in mAP@0.5:0.95 compared to the original model. The AP value for detecting craters, porosity, undercuts, and lack of fusion defects improves by 3.9%, 13.5%, 5.0%, and 2.5%, respectively. We conducted comparative experiments against other state-of-the-art models on the liquid reservoir weld dataset and the steel pipe weld defect dataset, and the results show that our model has outstanding detection performance.

## 1. Introduction

The liquid reservoir plays an important role in the automotive air conditioning system. Its primary function is to ensure that the refrigerant circulates in the appropriate state within the system. As shown in Figure 1, the liquid reservoir is typically assembled through welding to ensure airtightness. In recent years, automatic welding has become an important part of intelligent manufacturing for improving efficiency [1,2]. External variables in the welding process, such as inaccurate temperature regulation and inadequate surface cleaning of liquid reservoir, can lead to weld defects. The most commonly occurring types of defects are craters, porosity, undercuts, and lack of fusion, with representative samples shown in Figure 2. These defects directly undermine the sealing integrity of liquid reservoirs [3]. If a liquid reservoir with weld defects is installed, it can adversely affect the air conditioning system and potentially compromise vehicle safety [4,5,6]. Therefore, real-time monitoring and effective detection of weld defects after the liquid reservoir welding process is crucial [7].

In practical liquid reservoir welding, the frequency of defect occurrence is relatively low, yet multiple types of defects may coexist simultaneously. There is significant variation between different categories of defects, and even within the same defect type, the size and shape can be highly complex and variable. For example, crater defects are characterized by intricate shapes and considerable variations in area. Porosity defects tend to have smaller areas, making them susceptible to missed detection. In contrast, undercut defects and lack of fusion defects are narrow in shape, exhibit large length variations, and may even span the entire weld seam. These characteristics significantly complicate the task of defect detection. Manual inspection, though widely practised, is not only time-consuming and labor-intensive but also subject to the subjective judgment and work experience of the workers, resulting in unreliable results. Consequently, researchers have explored advanced techniques such as eddy current testing [8] and X-ray inspection [9] for detecting weld defects. However, these methods rely on physical equipment which is associated with high operational costs and is vulnerable to environmental factors. In the context of intelligent manufacturing and automated production, these challenges can be addressed by integrating advanced technologies such as deep learning and automated visual inspection systems [10,11,12].

Machine vision technology exhibits substantial potential for weld defect detection compared to the aforementioned non-destructive detection methods. Machine vision facilitates real-time image inspection while also offering comprehensive visual information regarding defects. Traditionally, weld defect detection is performed using image processing techniques combined with feature classification algorithms. Commonly employed feature classification algorithms include Support Vector Machines (SVMs) [13,14], Random Forest [15], and Decision Trees [16]. Malarvel et al. [9] introduced a weld defect detection and classification approach that employs the OTSU algorithm for image preprocessing, followed by an SVM for defect classification. Likewise, Hu et al. [17] proposed a weld defect detection methodology leveraging the BT-SVM classifier, which extracts defect attributes as classification features to identify weld defects. However, such methods encounter challenges in detecting complex shapes and small-sized defects. Furthermore, their performance is significantly influenced by environmental and lighting variations, leading to diminished adaptability and robustness.

In recent years, deep learning detection methods based on convolutional neural networks (CNNs) [18,19,20] have found important applications and developments. Representative networks such as VGGNet [21] and ResNet [22] are highly effective in automatic feature extraction and overcome the limitations of poor robustness inherent in traditional machine learning algorithms. Currently, object detection algorithms are generally categorized into two main types: two-stage and one-stage detection algorithms. Two-stage algorithms comprise R-CNN [23,24] and Faster R-CNN [25], while one-stage algorithms comprise YOLO [26,27,28] and SSD [29]. Numerous researchers have applied these algorithms to the field of weld defect detection. For instance, Liu et al. [30] proposed the AF-RCNN algorithm, which integrates ResNet and FPN as its backbone while employing an efficient convolutional attention module (ECAM). Similarly, Chen et al. [31] enhanced the Faster-RCNN algorithm by introducing the deep residual network Res2Net, which significantly improves feature extraction capabilities for weld defect detection.

However, these models enhance detection accuracy by increasing the number of parameters and model complexity, leading to excessive model size and limited inference speed. To address the challenge of low detection speed, Liu et al. [32] introduced an enhanced LF-YOLO algorithm. By incorporating an Efficient Feature Extraction (EFE) module, the model attained a detection speed of 61.5 FPS and an average precision of 92.9% on X-ray images of weld seam defects. To address overfitting due to the limited size of metal welding defect datasets, Li et al. [33] implemented Mosaic and Mixup augmentation techniques to augment the dataset. Additionally, they incorporated the lightweight GhostNet network to substitute the residual module in the CSP1 architecture, thereby reducing parameter count. The improved YOLOv5 algorithm adopted the Complete Intersection over Union (CIoU) loss function to improve bounding box regression precision. Wang et al. [34] introduced the YOLO-AFK model to handle complex welding defect scenarios. By integrating the Fusion Attention Network (FANet) and Alterable Kernel Convolution (AKConv) and designing a Cross-Stage Partial Network Fusion (C2f) module, they increased the model’s parameter count by just 12.4M while boosting accuracy, mAP, and FPS. In early 2023, Ultralytics launched the YOLOv8 model [35,36], delivering faster inference speeds, enhanced precision, and greater ease of training and adjustment.

Despite these advancements, deep learning models still face significant challenges in detecting liquid reservoir weld defects. The high variability in the shape and size of the same type of defect limits the generalization ability of these models, resulting in poor detection performance. To overcome these limitations, we propose an improved model based on YOLOv8s. In summary, this work makes the following contributions:

(1) The model’s neck is enhanced through the integration of the improved Reparameterized Generalized Feature Pyramid Network (RepGFPN) and an additional detection head with a size of 160 × 160. This significantly improves the model’s feature fusion capability while increasing its sensitivity to detecting small-area defects, such as porosity, as well as narrow defects like undercut defects and lack of fusion defects.

(2) The Focal Modulation Network (FocalNets) structure is incorporated into the YOLOv8s framework, replacing the Spatial Pyramid Pooling Fast (SPPF) module. This enhancement mitigates the degradation of detailed feature information, enhancing the model’s effectiveness in detecting complex defects within liquid reservoir weld seams.

(3) The Cascaded Group Attention (CGA) is incorporated within the C2f structure, effectively suppressing the propagation of redundant feature information in the neck and enabling more precise defect detection.

The remainder of this article is structured as follows: Section 2 presents the baseline, followed by a detailed discussion of the proposed improvement strategies and research methodologies. In Section 3, we describe the comparative experiments and ablation studies we conducted and present the results. Finally, Section 4 summarizes the conclusions and proposes potential directions for future research.

## 2. Model Introduction and Improvement

### 2.1. Basic YOLOv8 Model

YOLOv8 exhibits outstanding performance in defect detection, delivering superior accuracy and speed compared to its predecessors while remaining exceptionally easy to deploy. These characteristics make it suitable for real-time industrial applications. Therefore, we chose YOLOv8 as the baseline.

The YOLOv8 architecture consists of three main components—the backbone, neck, and head—as shown in Figure 3. The backbone handles feature extraction and includes the Conv (ConvBiSiLU), C2f (Faster Implementation of CSP Bottleneck with 2 convolutions), and SPPF modules. The C2f module, a lightweight convolutional structure, enhances gradient flow with cross-layer connections, thereby improving its feature representation capability. At the end of the backbone, the SPPF module enhances sensitivity by incorporating large-scale features, which boosts object detection performance. For the neck, a Feature Pyramid Network (FPN) [37] and Path Aggregation Network (PANet) [38] are utilized to integrate multi-scale features. The combination of the FPN-PANet structure and the C2f module aggregates feature maps from three different stages of the backbone, merging shallow information into deeper features. YOLOv8 introduces an anchor-free mechanism [39] to enhance the detection of objects with irregular heights and widths. YOLOv8 employs the TaskAlignedAssigner positive sample assignment strategy [40], which selects positive samples using a weighted scoring mechanism. Distribution Focal Loss [41] further focuses the model on hard-to-classify samples, addressing the imbalance between positive and negative samples in object detection tasks. To improve the overlap and localization accuracy between predicted and ground-truth bounding boxes, YOLOv8 incorporates CIoU Loss [42]. Furthermore, the detection heads transition from the coupled heads in YOLOv5 to the decoupled heads, which separates classification and regression tasks, theoretically enhancing training efficiency.

However, YOLOv8 still has limitations when detecting liquid reservoir weld seam defects. Specifically, the SPPF module tends to lose fine-grained feature information during pooling operations, thereby degrading the detection performance for intricate weld defects. Furthermore, the feature fusion mechanism in the original neck underperforms, adversely affecting both object localization and classification accuracy. As a result, YOLOv8 struggles to effectively detect defects with complex shapes and large size variations.

### 2.2. Improved YOLOv8 Model

To meet the requirements of real-time performance and accuracy in practical production inspections, we propose a defect detection method based on the YOLOv8 model. Compared with the other four models in the YOLOv8 series, YOLOv8s is distinguished by its optimal balance between performance and computational efficiency. It exhibits superior performance in real-time applications, especially in resource-limited environments. Consequently, we propose the following improvement strategies to YOLOv8s to improve detection precision. The architecture of the improved detection model is shown in Figure 4:

(1) The feature fusion component of the YOLOv8s neck is enhanced using the improved RepGFPN framework, with the addition of a 160 × 160-sized detection head.

(2) The original SPPF module in YOLOv8s is substituted with FocalNets.

(3) The CGA module is incorporated within the Bottleneck segment of the C2f module and linked to the decoupled heads.

#### 2.2.1. The Improved RepGFPN Structure

In YOLOv8, the FPN-PANet framework performs multi-scale feature fusion through top-down and bottom-up pathways. However, this approach inadequately integrates low-level fine-grained features with high-level global information, leading to feature loss or blurring that hinders accurate classification and localization of complex liquid reservoir weld defects.

To address this limitation, we incorporate an improved RepGFPN structure [42,43] into the YOLOv8s neck, replacing the original FPN-PANet. As illustrated in Figure 4, RepGFPN enables more effective multi-level feature fusion and extracts richer feature representations. It employs varying channel widths for different feature scales and utilizes the CSP (Cross-Stage Partial) structure [44] for reparameterization and hierarchical aggregation. This design combines features from adjacent (both upper and lower) and same-level layers while maintaining computational efficiency. By using concatenation instead of element-wise addition for feature fusion, we further mitigate information loss. Additionally, we introduce a 160 × 160 small-scale detection layer that provides high-resolution feature maps with enhanced detail, improving detection of fine defects such as small porosity, narrow undercuts, and lack of fusion.

Nevertheless, this enhancement may cause the loss of critical information from the 80 × 80, 40 × 40, and 20 × 20 detection heads during feature fusion. To alleviate this issue and reduce model complexity, we remove the PANet-based downsampling connections and disconnect the link between the 160 × 160 and 80 × 80 detection heads. While this adjustment preserves feature integrity and lowers computational cost, it may marginally affect detection accuracy. To compensate, we add an extra FPN structure after the FPN-PANet, enabling the 160 × 160 detection layer to effectively fuse information with other scales. This strengthens cross-scale feature propagation, enriches multi-level information flow, and improves detection performance for targets with large scale variations.

#### 2.2.2. Replacing SPPF with the Focal Modulation Networks

In fine-grained tasks, the SPPF module enhances global feature representation through multi-scale pooling aggregation. However, this process tends to compress edge information, textural details, and small-scale structures in liquid reservoir welds, compromising detection accuracy. These limitations become particularly evident when detecting weld defects with complex shapes, significant size variations, and numerous small targets. To address this issue, we replace the SPPF structure at the end of the YOLOv8 backbone with FocalNets [45]. The overall architecture is shown in Figure 5.

FocalNets employs a focal modulation mechanism to capture long-range dependencies and contextual information, formulated as follows:(1)$$y_{i}=q\left(x_{i}\right)⊙m\left(i,X\right)=q\left(x_{i}\right)⊙h\left(\sum_{\ell =1}^{L+1}g_{i}^{\ell }\cdot z_{i}^{\ell }\right)$$

In (Equation 1), the feature map $X\in R^{H\times W\times C}$ is projected via query function q(·) to generate visual tokens $x_{i}\in R^{C}$, where ⊙ denotes element-wise multiplication. The context aggregation function m(·) produces a modulator, which is multiplied with $x_{i}$ to yield $y_{i}\in R^{C}$. The modulator is derived as follows:(2)$$Z^{\ell }=f_{a}^{\ell }\left(Z^{\ell -1}\right)≜\mathrm{GeLU}\left(\mathrm{DWConv}\left(Z^{\ell -1}\right)\right)\in R^{H\times W\times C}$$(3)$$Z^{out}=\sum_{\ell =1}^{L+1}G^{\ell }⊙Z^{\ell }\in R^{H\times W\times C}$$

The input X is first projected to $Z^{0}=f_{z}\left(X\right)\in R^{H\times W\times C}$. Depthwise convolution (DWConv) and GeLU activation [46] then generate hierarchical representations $Z^{\ell }$ at each focal level ℓ∈{1,…,L}, as in (Equation 2). Global average pooling on the *L*-th level yields global context $Z^{L+1}=\mathrm{AvgPool}\left(Z^{L}\right)$, producing L+1 feature maps. Gating weights $G=f_{g}\left(X\right)\in R^{H\times W\times (L+1)}$ are obtained via a linear layer. The level-specific gating value $G^{\ell }\in R^{H\times W\times 1}$ is multiplied with $Z^{\ell }$ and summed to form $Z^{out}$ (Equation (Equation 3)). Finally, a linear layer processes $Z^{out}$ to generate the modulation map $M=h\left(Z^{out}\right)\in R^{H\times W\times C}$.

By focusing on key image regions, FocalNets mitigates feature compression caused by SPPF’s multi-level pooling. This approach effectively models cross-level contextual information, capturing finer details and enhancing recognition of complex weld defects.

#### 2.2.3. Adding the Cascaded Group Attention Modulation Network

The core idea of the improved RepGFPN model is to enhance multi-scale object detection through fusion of feature maps from different network layers. However, this approach introduces the challenge of redundant information flow across layers, which can degrade detection performance. To address this issue, we integrate the CGA module [47] into the feature fusion process of YOLOv8s.

The CGA module is embedded within the Bottleneck structure of the C2f module and connected to decoupled heads. This design enables effective multi-level feature fusion while reducing redundancy caused by cross-scale integration. The overall architecture is shown in Figure 6. The primary goal of CGA is to enhance feature diversity for attention heads, with the attention mechanism defined as follows:(4)$${\tilde{X}}_{ij}=Attn\left(X_{ij}W_{ij}^{Q},X_{ij}W_{ij}^{K},X_{ij}W_{ij}^{V}\right)$$(5)$$X_{ij}^{\prime }=X_{ij}+{\tilde{X}}_{i(j-1)},\;1<j\leq h$$(6)$${\tilde{X}}_{i+1}=Concat{\left[{\tilde{X}}_{ij}\right]}_{j=1:h}W_{i}^{P}$$

In (Equation 4), the CGA module partitions the input feature into multiple segments, each processed by a separate attention head. The *j*-th head computes self-attention on $X_{ij}$, where $X_{ij}$ represents the *j*-th partition of input feature $X_{i}=\left[X_{i1},X_{i2},\ldots ,X_{ih}\right]$, with 1≤j≤h and *h* being the total number of heads. The projection layers $W_{ij}^{Q}$, $W_{ij}^{K}$, and $W_{ij}^{V}$ map input features into distinct Query, Key, and Value subspaces, enabling richer feature extraction.

In (Equation 5), the input partition $X_{ij}$ is combined with the output from the previous head ${\tilde{X}}_{i(j-1)}$ to form $X_{ij}^{\prime }$, which then serves as an input for the *j*-th attention head. (Equation 6) shows how $W_{i}^{P}$ projects the concatenated outputs back to the original input dimension.

Through its cascading mechanism, CGA captures multi-level features and enhances feature interaction. Each head’s output progressively refines the next head’s input, improving feature representations. While the neck network’s multi-level feature extraction and fusion inevitably introduce redundancy into detection heads, CGA helps the model focus on critical features and suppresses redundant feature propagation, ultimately improving detection accuracy and robustness.

## 3. Experiments

### 3.1. Dataset

To evaluate the performance and generalization capability of the improved model, we conduct experiments on the liquid reservoir weld seam defect dataset discussed in this paper.

#### Dataset of Liquid Reservoir Welding Seam Defects

We construct a specialized dataset of welding seam defects, with sample images sourced from the liquid reservoir production line. A line laser profiler is used to scan liquid reservoir weld seams, obtaining three-dimensional contour data. The line laser profiler is fixed in position, and a rotating support mechanism comprising servo motors, gear shafts, and rollers is constructed. This system enables the line laser profiler to scan the weld seam on the uniformly rotating liquid reservoir, acquiring three-dimensional profile data at a scan speed of 3 s per unit. While these data provide precise geometric information, they are incompatible with mainstream detection models like YOLOv8. To address this, we transform the 3D point cloud into grayscale images by mapping depth information to pixel intensity, thereby visualizing the three-dimensional morphology in a two-dimensional representation. This conversion process requires an average processing time of 15–30 ms. This conversion not only reduces data dimensionality and enhances defect region visibility but also eliminates interference from color and texture variations common in traditional visual inspection, improving both robustness and interpretability. The resulting grayscale images can be directly utilized for training deep learning models, enabling efficient and accurate identification of weld defects without reliance on complex 3D data processing pipelines.

This dataset contains a total of 431 raw data samples, each with a resolution of 935 × 6000 pixels. These samples represent four distinct types of defects: crater defects, porosity defects, undercut defects, and lack of fusion defects. The original data samples are split into training and test sets in a 4:1 ratio, comprising 345 images for the training set and 86 images for the test set. For the minority defects in the training set, namely the porosity defects, undercut defects, and lack of fusion defects, various data augmentation techniques are applied. These include rotation, noise injection, adjustment of brightness and contrast, and Contrast-Limited Adaptive Histogram Equalization (CLAHE). These augmentation strategies help to balance the distribution of defect types within the training set, ensuring optimal model performance while enhancing its robustness and generalization capability and mitigating overfitting. After data augmentation, the training set is expanded to 681 images. A comparison of the defect count before and after data augmentation is shown in Figure 7a.

Furthermore, we normalize the width and length of defects and perform a statistical analysis. The scatter plot in Figure 7b illustrates the width and height of the defect bounding boxes. The deepest color region in the lower-left corner of the plot indicates a large number of defects with relatively small areas.Additionally, there is a distribution of scatter points along the upper and right sides of the plot, showing the presence of defect samples with relatively larger lengths or widths, indicating a large range of defect sizes within the dataset.

### 3.2. Experimental Environment and Training Strategies

The operating system used in this study was Ubuntu 24.04. We conducted all experiments on an Intel i5-14600KF CPU and a single NVIDIA RTX 4060Ti GPU with 16 GB of memory. All models were based on the deep learning framework PyTorch (version 1.10.0). The training parameter settings for the comparative experiment conducted on the liquid reservoir weld defect dataset are shown in Table 1.

### 3.3. Evaluation Metrics for Object Detection

In this experiment, the primary evaluation metrics are precision, recall, mean average precision (mAP), the number of parameters, and frames per second (FPS). Precision measures the model’s accuracy in predicting positive instances. Recall evaluates the model’s ability to identify positive instances. AP is the area under the precision–recall curve and provides a comprehensive assessment of the model’s performance, as defined in (Equation 7). mAP is the mean of the AP values across all categories, as defined in (Equation 8). Specifically, mAP@0.5 and mAP@0.5:0.95 denote the mean average precision calculated at different Intersection over Union (IoU) thresholds.

This study employs mAP@0.5 and mAP@0.5:0.95 as key evaluation metrics, comprehensively assessing the deep learning model’s detection accuracy and robustness from different perspectives. mAP@0.5, using an IoU threshold of 0.5, evaluates the model’s object recognition capability under relaxed localization requirements, focusing on presence detection accuracy. In contrast, mAP@0.5:0.95 computes the average precision across multiple IoU thresholds from 0.5 to 0.95 (step size 0.05), emphasizing precise localization performance. The combined use of these metrics enables a balanced evaluation of the model’s practical utility in general scenarios and its comprehensive performance under stricter criteria, thereby reducing potential biases from single-metric assessments.(7)$$AP=\int_{0}^{1}P\left(R\right)dR$$(8)$$mAP=\frac{1}{C}\sum_{i=1}^{C}AP_{i}$$

### 3.4. Experimental Results

#### 3.4.1. Comparison of Different Feature Fusion Structures

To validate the effectiveness of the proposed improved RepGFPN-based neck structure, we conduct comparative experiments. These experiments involve YOLOv8s, YOLOv8s-p2, YOLOv8s integrated with the BiFPN and AFPN structures, and YOLOv8s enhanced with the improved RepGFPN structure. All experiments were conducted under uniform training conditions, and the results are summarized in Table 2. In addition to the improved RepGFPN structure, the Bidirectional Feature Pyramid Network (BiFPN) [48] and Asymptotic Feature Pyramid Network (AFPN) [49] represent variants of Feature Pyramid Networks frequently utilized in object detection tasks. The BiFPN extends the original FPN by incorporating a bottom-up pathway, enabling bidirectional cross-scale connections to effectively utilize multi-scale features. The AFPN employs adaptive spatial fusion to dynamically adjust fusion weights according to the relative significance of the input features. By integrating high-level semantic information with low-level fine-grained details via cross-layer connections, these designs enhance feature representations.The Swin Transformer achieves adaptation to vision tasks by employing a hierarchical feature map architecture, inspired by CNNs, and computing self-attention within shifted local windows. This design retains the powerful global modeling capacity of Transformers while maintaining linear computational complexity, making it a versatile and efficient backbone for a wide range of vision tasks, including image classification, object detection, and segmentation. The structures of improved the RepGFPN, BiFPN, AFPN, and Swin Transformer are frequently employed to optimize the neck of deep learning models, thereby substantially improving detection performance.

The comparative results indicate that YOLOv8s-p2, which integrates an additional 160 × 160 detection layer into the YOLOv8s, outperforms the baseline, achieving improvements of 1.4% and 0.3% in mAP@0.5 and mAP@0.5:0.95, respectively. Replacing the FPN-PANet in YOLOv8s-p2 with the improved RepGFPN structure achieves an additional 2.7% improvement in mAP@0.5 and a 0.9% improvement in mAP@0.5:0.95 compared to YOLOv8s-p2. Although both BiFPN and AFPN effectively reduce model complexity, the YOLOv8s-p2 model incorporating the improved RepGFPN structure achieves superior performance in both mAP@0.5 and mAP@0.5:0.95. While BiFPN enhances multi-scale feature interaction through weighted bidirectional fusion, its repeated cross-scale connections may retain redundant feature information. AFPN mitigates feature disparities between non-adjacent layers via progressive fusion yet exhibits limited efficiency in deep semantic information propagation. Although the Swin transformer models global dependencies through shifted window mechanisms, its excessive parameter count often leads to optimization challenges and loss of fine-grained features. In contrast, the improved RepGFPN strengthens multi-scale feature representation while maintaining efficient gradient flow through structural reparameterization and streamlined connections, thereby achieving a better balance between detection accuracy and inference speed.

#### 3.4.2. Comparison of FocalNets with Varying Numbers of Convolutional Layers

In this paper, we introduce FocalNets modules with varying numbers of convolutional layers to enhance the baseline. As the number of convolutional layers changes, the sizes of the convolutional kernels at each layer are adjusted accordingly while maintaining a consistent small stride. For simplicity, the kernel size of each layer is $k^{\ell }\times k^{\ell }$, with the first layer having a kernel size of 3 × 3. We gradually increase the kernel size from lower focal levels to higher ones, increasing by 2 at each step, so the kernel size of the *ℓ*-th layer is $k^{\ell }=k^{\ell -1}+2$. Through experimental evaluation, we observe that the model achieves optimal detection performance when the number of convolutional layers is set to 2, as shown in Table 3. The model’s mAP@0.5 and mAP@0.5:0.95 is increased by 2.7% and 2.3%, respectively.

#### 3.4.3. Comparison of Different Attention Modules

We compare four attention mechanisms—CBAM, ECA, SE, and CGA—to further explore their efficacy in reducing redundant information propagation and enhancing the model’s detection performance.

CBAM is a module that integrates both channel attention and spatial attention, enabling the extraction of image features by focusing on distinct channel and spatial locations. ECA, an efficient and computationally lightweight channel attention mechanism, captures local dependencies between channels through localized convolutions, thereby emphasizing important channel features without incurring substantial computational overhead. SE is a mechanism that enhances the model’s representational power by adaptively adjusting the feature responses across channels. It allows the model to automatically learn which channels are critical for the task by dynamically modifying channel feature importance. The CGA attention mechanism selectively enhances useful details in target locations and feature channels, allowing the model to focus more on crucial regions. The comparative results are presented in Table 4. The CGA module enables the model to achieve the maximum mAP@0.5, outperforming other attention mechanisms. Compared to the baseline, mAP@0.5 and mAP@0.5:0.95 are increased by 2.2% and 0.2%, respectively.

#### 3.4.4. Ablation Experiments

As shown in Table 5, after incorporating the improved RepGFPN structure into the neck of the baseline, the mAP@0.5 and mAP@0.5:0.95 increase by 4.1% and 1.2%, respectively. This demonstrates that the improved neck structure effectively integrates multi-scale feature information, significantly enhancing the model’s ability to detect defects with large dimensional variations. The FocalNets, by extracting rich contextual information and employing a focal modulation mechanism, improves the model’s detection accuracy for complex defects. The connection of the detection head with the C2f structure integrated with the CGA module effectively addresses the issue of redundant information transmission during feature fusion in the neck. This enables the model to focus on more relevant feature information, thereby improving performance. When both the FocalNets module and the improved RepGFPN structure are added simultaneously, the performance improvement is more significant than when added individually. After incorporating all the enhancement modules, the improved model demonstrates an increase of 6.3% in mAP@0.5 and 4.3% in mAP@0.5:0.95 compared to the baseline. These results indicate that the addition of the CGA module addresses the issue of redundant information transmission between different levels in the improved RepGFPN structure, further enhancing the model’s detection performance.

However, the increased network depth and parameter count reduced the frame rate (FPS) of the final improved model, highlighting the inherent trade-off between precision (mAP) and processing speed. Nevertheless, the total time required for a complete inspection of a single liquid reservoir weld—comprising image acquisition by the line laser profiler (approximately 3 s), 3D point cloud to grayscale image conversion (averaging 15–30 ms), and image inference by the enhanced model—still satisfies the production cycle requirement of 5 s per unit.

To evaluate the improvement in detection performance across four defect categories, we compare the improved model with YOLOv8s. Figure 8 shows the mAP@0.5 values for each defect category in both the baseline and improved models. The enhanced model demonstrates significant improvements in detecting crater defects, porosity defects, undercut defects, and lack of fusion defects, with AP values increasing by 3.9%, 13.5%, 5.0%, and 2.5%, respectively. The significant improvement in the porosity detection performance of the improved model can be attributed to the incorporation of a 160 × 160 small-sized detection layer, which enables the model to better identify minute defects. The integration of the improved RepGFPN structure enhances the model’s ability to detect craters with complex shapes, as well as undercuts and lack of fusion with varying aspect ratios. FocalNets captures richer detailed feature information, further boosting the model’s capability in identifying intricate defects. Moreover, the combination of the improved RepGFPN with the CGA attention mechanism effectively suppresses the transmission of redundant information, thereby improving the overall detection performance. These advancements enable the model to conduct more precise inspections while enhancing its ability to detect defects with complex shapes and varying scales.

However, the relatively low AP values achieved by the improved model in detecting crater and porosity defects indicate persistent limitations. Crater defects exhibit high diversity in shape, size, and depth, while their low contrast and irregular morphology undermine feature extraction effectiveness, making it challenging for the improved model to fully capture their polymorphic characteristics. Although porosity defects possess relatively distinct circular or elliptical structures that facilitate spatial pattern recognition by the enhanced RepGFPN architecture, both the quantity and diversity of training samples for this defect category are substantially limited compared to other classes. This data distribution imbalance restricts sufficient representation learning, thereby hindering further breakthrough in detection performance.

#### 3.4.5. Comparison of Different Detection Models

To further demonstrate the effectiveness of the proposed improved model, we conducted comparative experiments with several other mainstream detection models and existing weld seam defect detection methods. All experiments were conducted using the liquid receiver weld defect dataset.

The results of the comparative experiments are shown in Table 6. RT-DETR (Real-Time DEtection TRansformers) is a Transformer-based model optimized for real-time object detection tasks. It builds upon the DETR (Detection Transformer) framework, incorporating optimizations and adjustments. However, it has a relatively large number of parameters and suboptimal detection performance. As more advanced versions in the YOLO series, YOLOv9s and YOLOv10s demonstrate excellent detection performance in industrial scenarios. Although the larger number of parameters results in slower detection speeds, they provide significant improvements in accuracy and robustness. The LF-YOLO model [13] and the YOLOv5 with GhostNet architecture and CIoU [14] demonstrate impressive detection performance and speed while maintaining a lower parameter count. On the other hand, the YOLO-AFK model [21] has an excessively large parameter count and performs poorly on the liquid reservoir weld defect dataset. In our comparative experiment, the improved model outperformed others in terms of both mAP@0.5 and mAP@0.5:0.95, achieving the highest detection performance.

#### 3.4.6. Visualization Analysis of Improved Model

To provide a clearer and more intuitive analysis of the impact of the improvement modules on detection, we employ the gradient-weighted class activation mapping (Grad-CAM) technique [50] for visual analysis. This approach helps explain the model’s prediction results by illustrating the decision-making process and highlighting the regions the model focuses on during detection, thereby enhancing the model’s interpretability. Grad-CAM generates visual explanations for the decisions made by deep neural networks, helping to understand which regions of the image the model focuses on when making predictions.

Figure 9 shows the original weld defect images, along with the heatmaps generated by applying Grad-CAM to YOLOv8s, YOLOv8s-improved RepGFPN, and the final improved model. It can be observed that the baseline model exhibits a weak response and scattered attention to defect regions, indicating insufficient recognition capability for target defects. Compared to the baseline model (b), the heatmap of the improved RepGFPN model (c) demonstrates higher attention concentration on defects. Although this model can initially localize defects, its attention coverage and response intensity remain noticeably weaker than those of the complete model (d). The heatmaps generated by the complete model display a more focused and coherent attention distribution, reflecting a significant enhancement in both focusing intensity and localization accuracy on defect regions, which strongly validates the effectiveness of the proposed improvement strategy.

Figure 10 shows the detection results for sample images from different models. From the original sample images, it is obvious that there are significant variations in the shapes and sizes of the craters, the porosity defects exhibit a relatively small size, and the aspect ratios of the undercut defects and lack of fusion defects show significant disparity. Additionally, these defects can occur simultaneously. These factors result in the tendency of YOLOv8s to generate false alarms and missed detections when detecting weld defects. In comparison, the proposed improved model demonstrates superior accuracy in detecting the target defects. These visual analyses validate that the improved model has superior detection capability for liquid reservoir weld defects.

#### 3.4.7. Dataset of Steel Pipe Weld Defects

To validate the generalization capability and robustness of the improved model, we acquired a publicly available dataset of steel pipe weld defects for detection purposes. This selection is justified by the following three reasons. First, weld seams in steel pipes and liquid reservoirs exhibit high similarity in defect types, which ensures the relevance of the validation. Second, although the components differ, the data modalities of the two datasets—such as grayscale images and height maps—are fundamentally consistent. This consistency enables the model to learn generic defect features independent of the specific component. Finally, the use of a large-scale public dataset effectively mitigates the issue of data scarcity in specific scenarios, thereby allowing for a more rigorous and objective evaluation of the model’s generalization capability across diverse defect patterns.

The dataset encompasses eight types of defects represented in X-ray imaging—air-holes, bite edges, broken arcs, cracks, overlapping, slag inclusion, lack of fusion, and hollow beads—totaling 3408 images. The specific sample sizes for each defect category and the corresponding actual samples are shown in Table 7 and Figure 11. Since the model achieves satisfactory training results with this dataset, we do not apply data augmentation to it. Similarly, the dataset is divided in a 4:1 ratio, resulting in a training set of 2726 images and a test set of 682 images.

We employed various models in comparative experiments to evaluate the performance and generalization capability of the proposed improved model. The training epochs were set to 250, while all other parameters remained consistent with those specified in Table 2.

By comparing the detection performance of the proposed model with that of other advanced models, it is evident that the improved model achieves commendable detection results, with recall, mAP@0.5, and mAP@0.5:0.95 all reaching their maximum values. Furthermore, the improved model demonstrates a significant enhancement compared to the baseline. The results of the comparative experiments and the detection performance of the improved model are presented in Table 8 and Figure 12, respectively.

## 4. Conclusions

This paper proposes an improved YOLOv8s-based deep learning method for detecting liquid reservoir weld defects. To enhance the detection performance of YOLOv8s, a series of improvement strategies are implemented: first, the original neck network is optimized by introducing an improved RepGFPN feature fusion structure and adding a small-target detection head with a scale of 160 × 160; second, the SPPF module is replaced with the FocalNets module; finally, a C2f structure integrated with the CGA module is connected before the detection heads. Systematic ablation and comparative experiments demonstrate that the enhanced model achieves increases of 6.3% in mAP@0.5 and 4.3% in mAP@0.5:0.95 over the baseline. For specific defects, AP values improve by 3.9% for craters, 13.5% for porosity, 5.0% for undercuts, and 2.5% for lack of fusion, confirming the effectiveness of our modifications.

Future work will focus on extending the applicability boundaries of the proposed improved model by systematically validating its defect detection capabilities for industrial components made of diverse materials, such as metal alloys, engineering plastics, and composite materials. We will construct a cross-material sample dataset to investigate the model’s feature generalization performance on heterogeneous material surfaces. Furthermore, the multi-scale feature fusion mechanism will be optimized to account for differences in textual characteristics and defect manifestations across materials, thereby enhancing the model’s adaptability and practical utility in complex industrial environments.

## Figures and Tables

**Figure 1 sensors-25-06521-f001:**
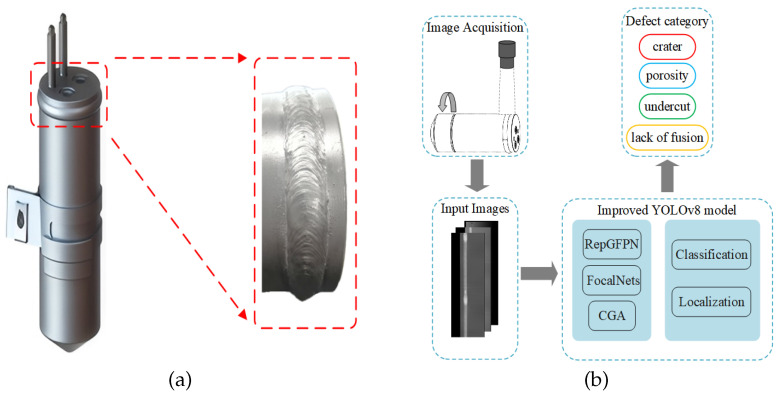
(**a**) Reservoir welding seam; (**b**) overall framework of the liquid reservoir weld defect detection method.

**Figure 2 sensors-25-06521-f002:**
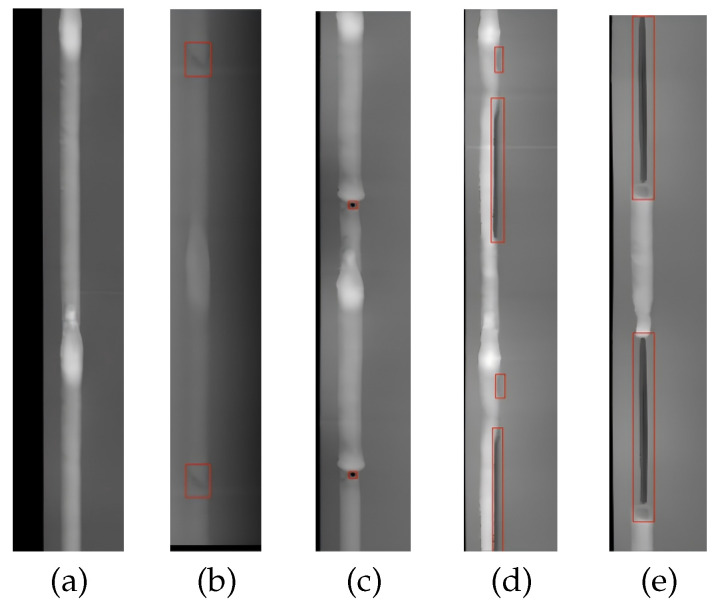
Examples of a normal weld and four types of weld defects. (**a**) normal weld; (**b**) craters; (**c**) porosity; (**d**) undercutting; (**e**) lack of fusion.

**Figure 3 sensors-25-06521-f003:**
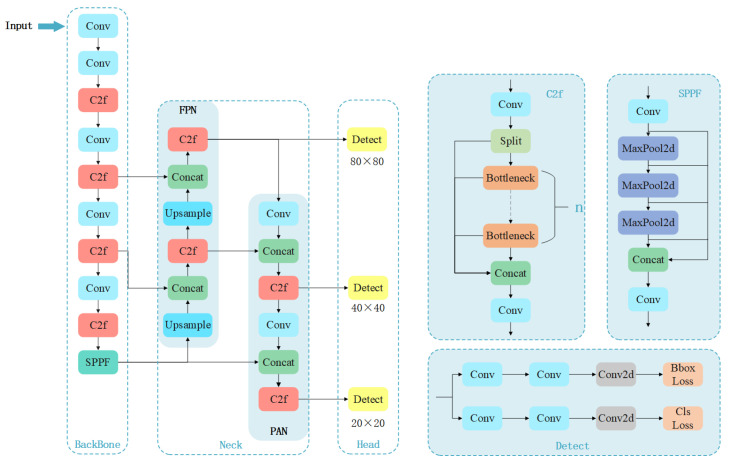
The structure of YOLOv8.

**Figure 4 sensors-25-06521-f004:**
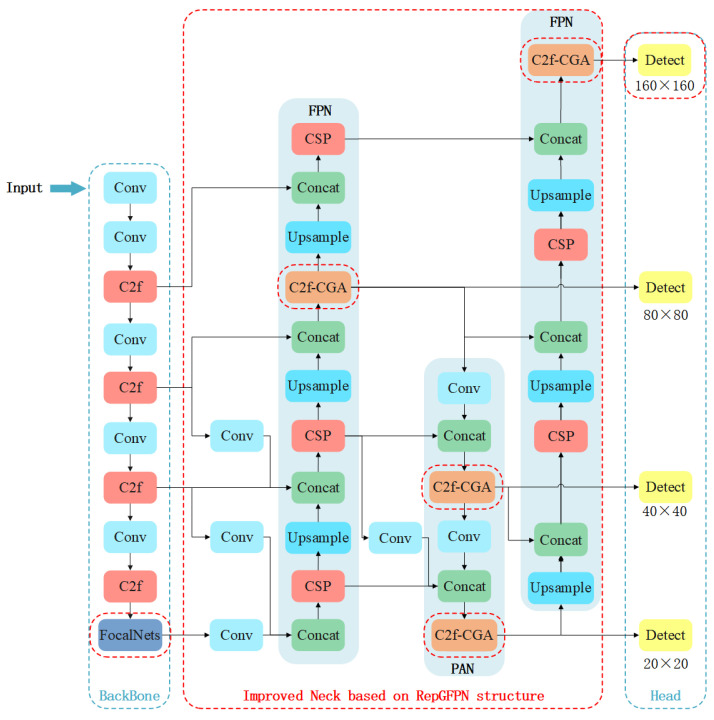
The structure of the improved YOLOv8s.

**Figure 5 sensors-25-06521-f005:**
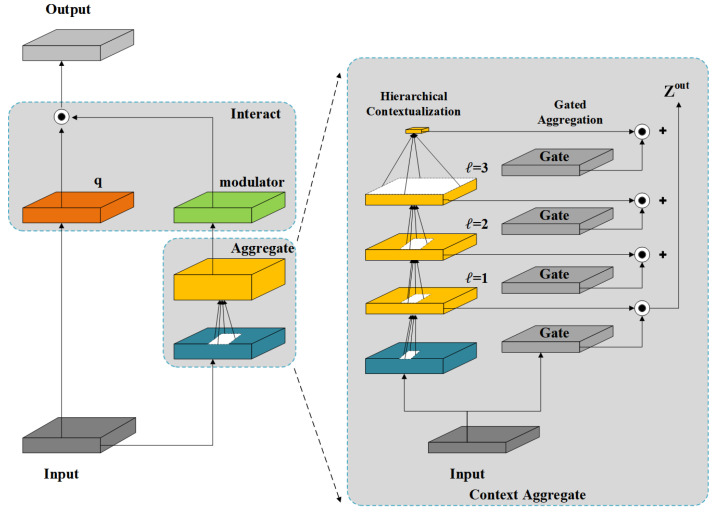
Architecture of FocalNets.

**Figure 6 sensors-25-06521-f006:**
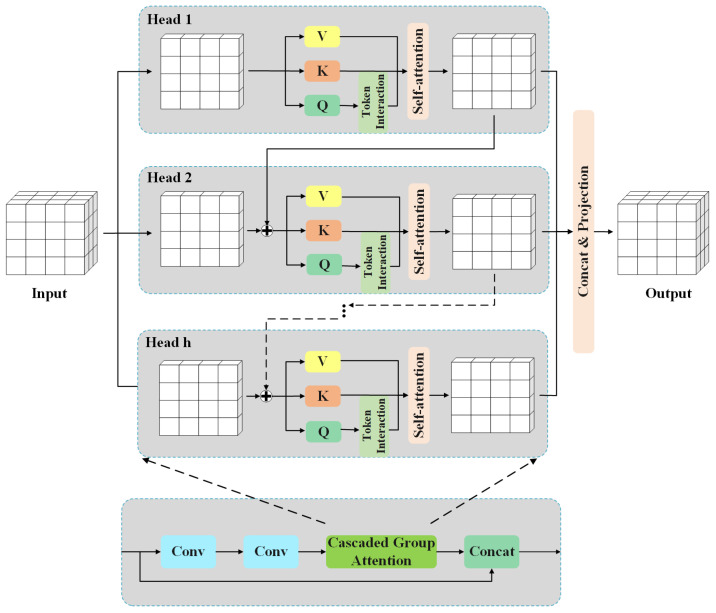
Structure of Cascaded Group Attention.

**Figure 7 sensors-25-06521-f007:**
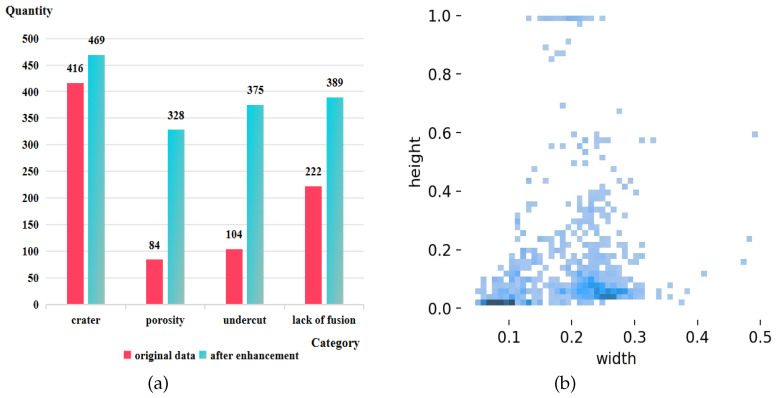
(**a**) Comparison of defect count before and after data augmentation on the liquid reservoir weld defect dataset; (**b**) size distribution of bounding boxes.

**Figure 8 sensors-25-06521-f008:**
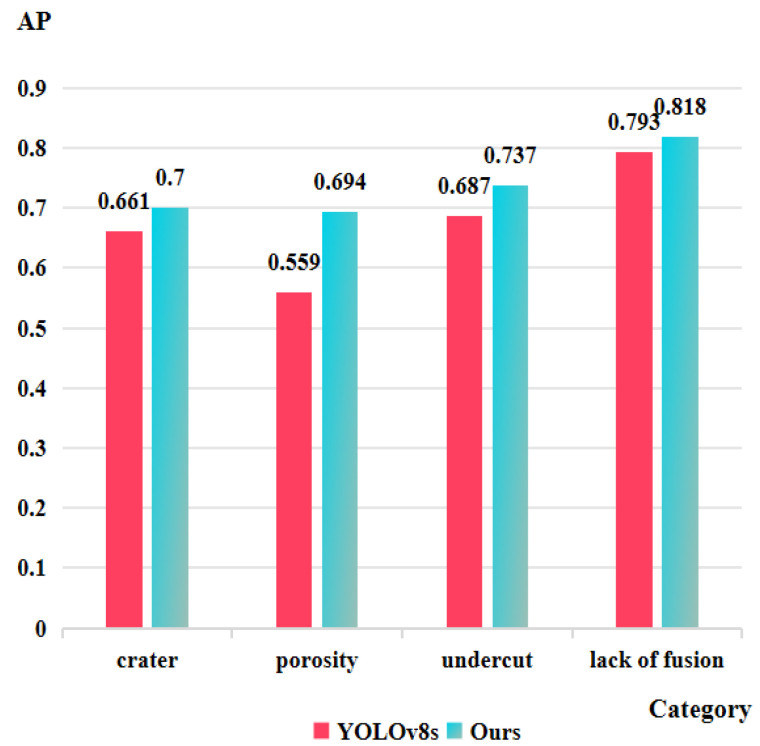
Comparison of AP values for different defect categories on the liquid reservoir weld defect dataset.

**Figure 9 sensors-25-06521-f009:**
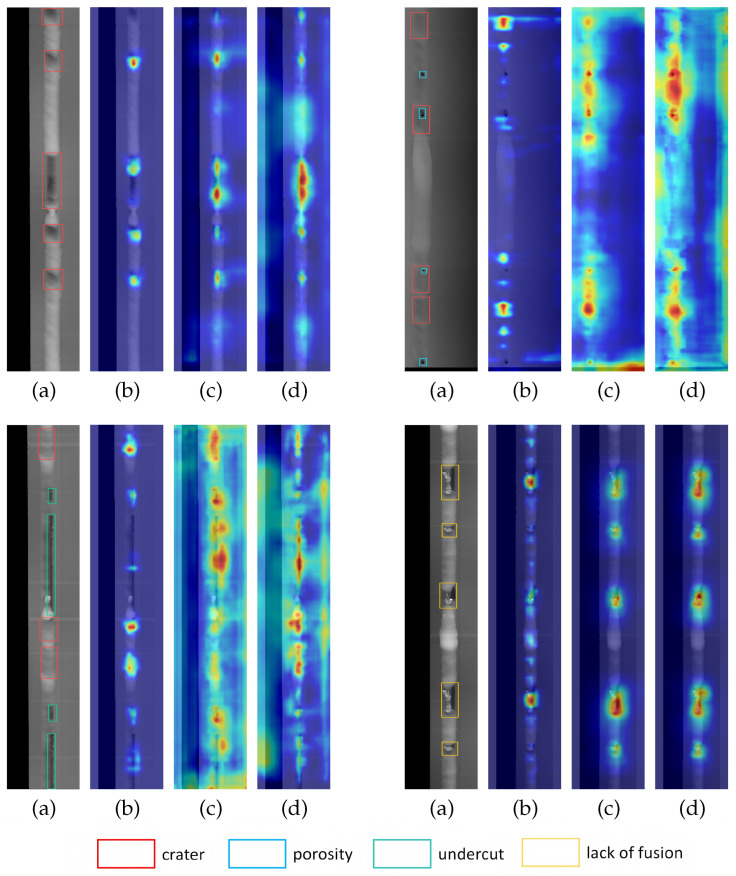
Comparison of heatmaps generated by different models on the liquid reservoir weld defect dataset. (**a**) Ground truth; (**b**) YOLOv8s; (**c**) YOLOv8s-improved RepGFPN; (**d**) YOLOv8s-improved RepGFPN-FocalNets-CGA.

**Figure 10 sensors-25-06521-f010:**
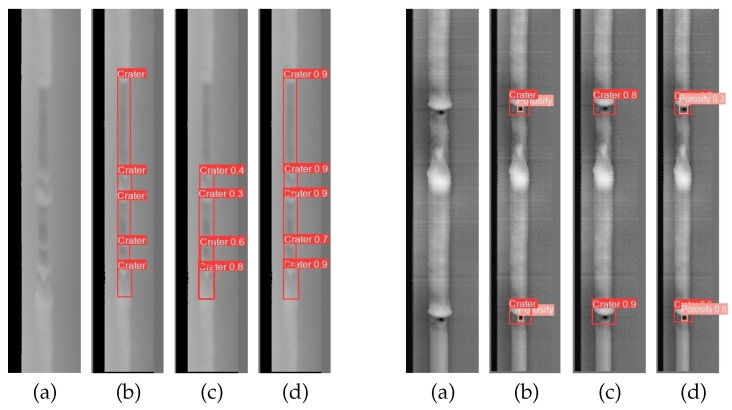
Comparison of the detection results of different models on the liquid reservoir weld defect dataset. (**a**) Original images; (**b**) ground truth; (**c**) YOLOv8s; (**d**) YOLOv8s-improved RepGFPN-FocalNets-CGA.

**Figure 11 sensors-25-06521-f011:**
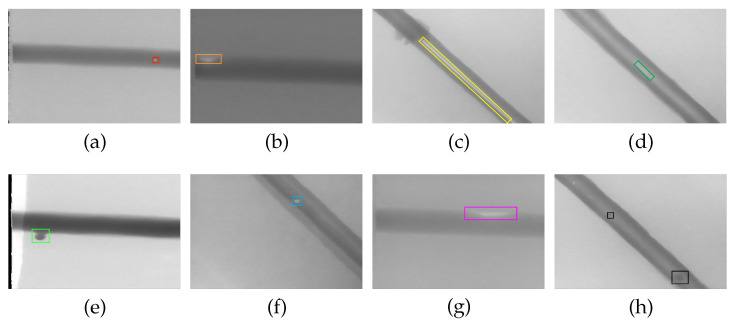
Examples of a normal weld and eight weld defects on the steel pipe weld defect dataset. (**a**) air-hole; (**b**) bite edges; (**c**) broken arc defect; (**d**) cracking; (**e**) overlapping; (**f**) slag inclusion; (**g**) lack of fusion; (**h**) hollow beads.

**Figure 12 sensors-25-06521-f012:**
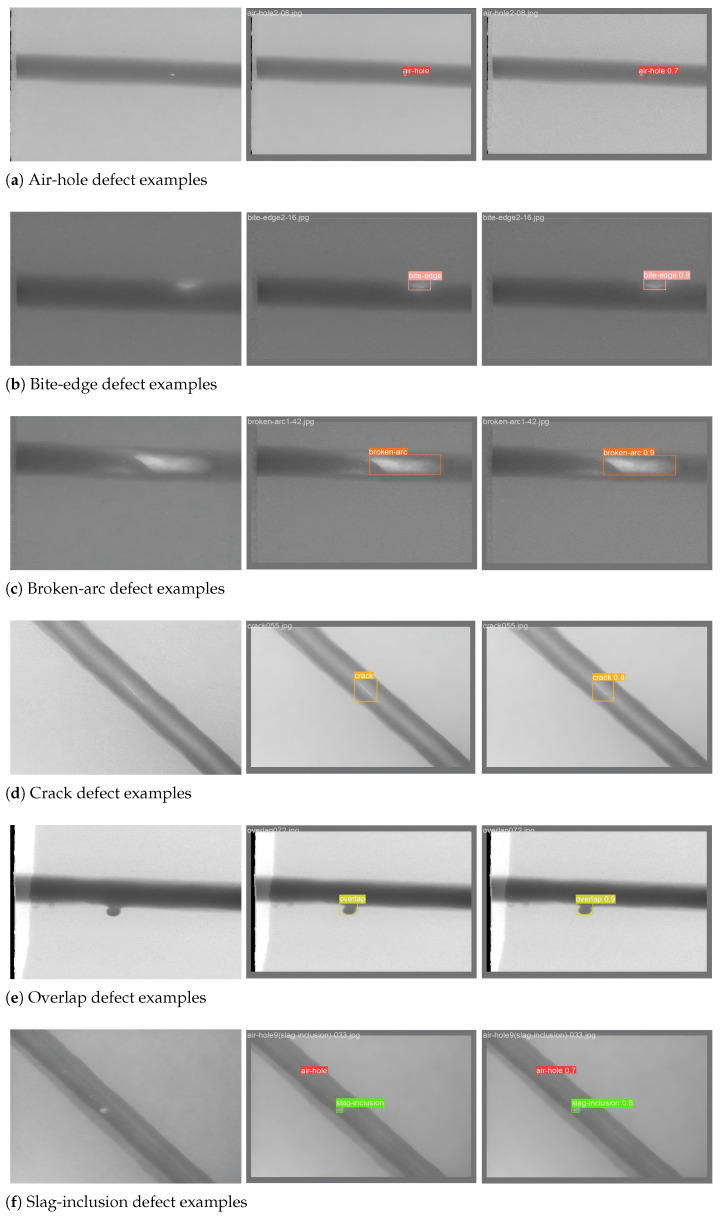
Representative detection results of the improved model on the steel pipe weld defect dataset. First column: original images. Second column: ground truth. Third column: YOLOv8s-improved RepGFPN-FocalNets-CGA.

**Table 1 sensors-25-06521-t001:** Experimental environment configuration.

Parameters	Setup
Epoch	150
Momentum	0.937
Initial learning rate	0.01
Final learning rate	0.001
Weight decay	0.0005
Batch size	16
Input image size	640 × 640
Optimizer	SGD
Patience	50

**Table 2 sensors-25-06521-t002:** Comparison of different necks on the liquid reservoir weld defect dataset.

Models	Precision/%	Recall/%	mAP@0.5/%	mAP@0.5:0.95/%	Params/M	FPS
YOLOv8s	76.6	65.0	67.5	33.2	11.13	337
YOLOv8s-p2	75.9	66.1	68.9	33.5	10.62	316
YOLOv8s-BiFPN	79.5	58.8	68.0	31.3	8.11	251
YOLOv8s-AFPN	71.6	63.2	69.2	32.5	6.71	222
YOLOv8s-Swin transformer	71.0	58.3	65.4	30.2	34.70	169
YOLOv8s-improved RepGFPN	79.0	67.9	71.6	34.4	13.34	231

**Table 3 sensors-25-06521-t003:** Comparison of FocalNets with different focus layer numbers on the liquid reservoir weld defect dataset.

Number of Focal Levels	Precision/%	Recall/%	mAP@0.5/%	mAP@0.5:0.95/%
Baseline	76.6	65.0	67.5	33.2
1	74.7	66.7	69.2	33.7
2	75.0	66.3	70.2	35.5
3	72.1	59.4	67.0	33.3
4	83.8	57.2	69.6	35.0

**Table 4 sensors-25-06521-t004:** Comparison of different attention mechanisms on the liquid reservoir weld defect dataset.

Models	Precision/%	Recall/%	mAP@0.5/%	mAP@0.5:0.95/%
Baseline	76.6	65.0	67.5	33.2
CBAM	72.4	68.9	68.9	34.7
ECA	74.9	65.7	69.1	34.1
SE	80.1	65.7	68.3	33.3
CGA	75.5	67.0	69.7	33.4

**Table 5 sensors-25-06521-t005:** Results of ablation experiments on the reservoir weld defect dataset.

Improved RepGFPN	FocalNets	CGA	Precision/%	Recall/%	mAP@0.5/%	mAP@0.5:0.95/%	Params/M	FPS
			76.6	65.0	67.5	33.2	11.13	337
✓			79.0	67.9	71.6	34.4	13.34	231
	✓		75.0	66.3	70.2	35.5	11.54	297
		✓	75.5	67.0	69.7	33.4	11.25	300
✓	✓		78.7	63.1	72.4	36.5	13.74	226
✓	✓	✓	79.2	66.6	73.8	37.5	13.96	198

**Table 6 sensors-25-06521-t006:** Comparative experiments of different models on the reservoir weld defect dataset.

Models	Precision/%	Recall/%	mAP@0.5/%	mAP@0.5:0.95/%	Params/M	FPS
RT-DETR	70.2	59.9	63.9	32.3	31.99	50
YOLOv8s	76.6	65.0	67.5	33.2	11.13	337
YOLOv9s	78.5	63.8	69.1	31.7	9.60	80
YOLOv10s	85.0	65.8	72.2	34.7	8.04	94
LF-YOLO	74.5	66.4	66.9	28.3	7.25	217
GC-YOLO	75.8	66.3	70.2	30.8	3.68	263
YOLO-AFK	74.3	66.1	69.9	33.7	63.14	62
Ours	79.2	66.6	73.8	37.5	13.96	198

**Table 7 sensors-25-06521-t007:** The number and proportion of images for each defect category on the steel pipe weld defect dataset.

Category	Air-Hole	Bite Edge	Broken Arc	Crack	Overlap	Slag Inclusion	Lack of Fusion	Hollow Bead
Number	1339	35	531	119	219	136	416	613

**Table 8 sensors-25-06521-t008:** Comparative experiments of different models on the steel pipe weld defect dataset.

Models	Precision/%	Recall/%	mAP@0.5/%	mAP@0.5:0.95/%	Params/M	FPS
RT-DETR	85.7	90.6	95.1	64.5	31.99	50
YOLOv8s	95.5	94.7	98.5	70.8	11.13	281
YOLOv9s	97.3	97.2	98.8	74.1	9.60	165
YOLOv10s	96.4	97.2	98.9	71.3	8.04	238
LF-YOLO	97.3	97.7	98.9	68.3	7.25	128
GC-YOLO	97.3	97.7	98.9	68.3	3.68	174
YOLO-AFK	96.4	96.3	98.7	72.1	63.14	31
Ours	96.7	97.9	98.9	74.6	13.96	110

## Data Availability

Data are not publicly available but can be obtained by contacting the corresponding author if necessary.

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
