# Peer review of "Liquid Reservoir Weld Defect Detection Based on Improved YOLOv8s"

_sensors, 2025, doi:10.3390/s25216521_

Round 1

Reviewer 1 Report

Comments and Suggestions for Authors

This article presents an improved YOLOv8-based detection model for identifying weld defects in liquid reservoirs. The paper addresses an industrially relevant problem and demonstrates solid experimental work. The overall structure of the paper is clear, logically organized, and easy to follow. The contribution is well described but somewhat incremental relative to existing YOLO-based frameworks. The following comments are provided for the authors’ further consideration:

  1. As described in Section 3.1.1 (“Dataset of liquid reservoir welding seam defects”), the dataset contains only 431 samples. This small sample size raises concerns regarding the generalization and robustness of the model. Although the authors employ data augmentation techniques to improve data diversity and categorical balance, the absolute number of samples remains limited compared to the number of network parameters. It would strengthen the paper to include additional validation results—such as cross-validation or testing on larger dataset—to make the claimed performance improvements stronger.
  2. The authors should better justify using RepGFPN over other widely adopted feature fusion strategies such as BiFPN (EfficientDet), or Transformer-based necks (e.g., Swin-transformer). A short comparison or rationale explaining why RepGFPN was preferred.
  3. The heatmaps in Figure 10 are informative but lack a clear scale or color legend. Including one would help readers better interpret the model’s focus regions and attention behavior.

Author Response

Comments 1:As described in Section 3.1.1 (“Dataset of liquid reservoir welding seam defects”), the dataset contains only 431 samples. This small sample size raises concerns regarding the generalization and robustness of the model. Although the authors employ data augmentation techniques to improve data diversity and categorical balance, the absolute number of samples remains limited compared to the number of network parameters. It would strengthen the paper to include additional validation results—such as cross-validation or testing on larger dataset—to make the claimed performance improvements stronger.

Response 1:Agree. Thank you for your valuable feedback. We acknowledge that the weld defect dataset of the liquid reservoir is limited in scale and may affect the validation of the model's generalization ability and robustness. To address this, we have introduced a publicly available dataset of steel pipe weld defects as a supplement. This dataset includes 3,408 X-ray images covering eight types of defects. The dataset is split into a training set (2,726 images) and a test set (682 images) in a 4:1 ratio. Comparative experiments with several mainstream models under the same experimental settings demonstrate that our proposed improved model still achieves excellent performance even on this larger dataset, further validating its generalization capability. The relevant content has been added to Section 3.4.7 on page 18.

Comments 2:The authors should better justify using RepGFPN over other widely adopted feature fusion strategies such as BiFPN (EfficientDet), or Transformer-based necks (e.g., Swin-transformer). A short comparison or rationale explaining why RepGFPN was preferred.

Response 2:Thank you for pointing out the limitations in our experiments. We have supplemented relevant experiments in Section 3.4.1 on page 11. To better justify our choice of RepGFPN as the feature fusion strategy, we designed a comparative study in that section, comparing the improved model against several mainstream feature fusion strategies—including YOLOv8s-p2 with BiFPN, AFPN, and Swin transformer. The experimental results show that RepGFPN achieves better performance across multiple evaluation metrics and more effectively balances detection accuracy and inference speed, thereby validating its suitability and advantage for the weld defect detection task.

Comments 3:The heatmaps in Figure 10 are informative but lack a clear scale or color legend. Including one would help readers better interpret the model’s focus regions and attention behavior.

Response 3:Agree. Thank you for your suggestion regarding the figures. Following your recommendation, we have added a corresponding color legend to the heatmap in Figure 10 (now Figure 9 on Page 16). The legend clearly indicates the defect type represented by each colored bounding box, helping readers better understand the model's attention intensity and distribution across different defect regions.

Reviewer 2 Report

Comments and Suggestions for Authors

Dear Authors, 

You have proposed an improved Yolo architecture to detect weld defects in liquid reservoirs of AC. Specifically,  the paper proposes an imporved YOLOv8s architecture,  which utlise  RepGFPN, FocalNets and CGA, to enhance the detection accuracy ( mAP@.5)

Your introduction and the model improvement,  is very well written. This section clearly justify the reason behind each modification. 

Following are comments to improve the manuscript. 

  1. Kindly discuss how you generate the weld defects images. Do you capture  x-ray images? Mention this under 3.1.1.  Also, why steel pipe weld defects dataset? Discuss this under 3.1.2
  2. Under 3.3, discuss which metric is the key for performance analysis. In results section, your data presents that some models exhibit superiority in terms of precision score while your model performance well in terms of mAP@.5
  3. Figure 9: state that this is for liquid reservoirs dataset.
  4. As of figure 10, isn't YOLOv8s with RepGFPN sufficient? When comparing images (c) and (d) in Figure 10, the results obtained by (c) is sufficiently accurate.  Also, by incorporating RepGFPN-FocalNets-CGA into YOLOv8s, you are increasing the system complexity. Please comment on this.
  5. Improve the conclusion by adding statements on performance imporvement achieved. It is advised to add numerical values.
  6. Line 274 should be "Fig 7(a)" not Fig 6 (a).

Author Response

Comments 1:Kindly discuss how you generate the weld defects images. Do you capture  x-ray images? Mention this under 3.1.1.  Also, why steel pipe weld defects dataset? Discuss this under 3.1.2.

Response 1:Agree. Thank you for your valuable feedback. We have made the following additions to the manuscript as suggested:

In lines 245-260 on Page 9, we have added a description of how the liquid reservoir weld defect images were generated: In this study, a fixed line laser profiler scans the weld seam on a uniformly rotating liquid reservoir via a servo-driven mechanism, acquiring 3D contour data at 3 seconds per unit. As this data format is incompatible with standard detection models like YOLOv8, we transform the 3D point cloud into grayscale images by mapping depth to pixel intensity, a process that averages 15–30 ms. This conversion not only reduces data dimensionality and enhances defect visibility but also eliminates interference from color and texture variations. The resulting images enable direct training of deep learning models for efficient weld defect identification, bypassing complex 3D processing.

In lines 469-478 on Page 18, we further elaborate on the rationale for introducing the steel pipe weld defect dataset. The defect types in steel pipe and liquid reservoir welds are highly similar, both covering typical defects, ensuring the validity of the comparison. Although the component shapes differ, the data modality remains consistent, as both are morphology-based grayscale images. This consistency helps the model learn component-agnostic general features. Moreover, this large-scale public dataset alleviates the issue of limited sample size and enables a more comprehensive evaluation of the model’s generalization capability across diverse defect patterns.

Comments 2:Under 3.3, discuss which metric is the key for performance analysis. In results section, your data presents that some models exhibit superiority in terms of precision score while your model performance well in terms of mAP@0.5.

Response 2:Agree. Thank you for your valuable feedback on this matter. In response to your suggestion, we have added a clarification of the key performance evaluation metrics in Section 3.3 on Page 10. For the weld defect detection task in this study, we select mAP@0.5 and mAP@0.5:0.95 as the primary metrics for assessing model performance. By combining these two types of metrics, we are able to evaluate both the practical performance of the model in general industrial scenarios and its comprehensive performance under strict localization criteria. This approach helps prevent potential evaluation bias that could arise from relying on a single metric, thereby providing a more thorough basis for judging the model's reliability in real-world applications.

Comments 3:Figure 9: state that this is for liquid reservoirs dataset.

Response 3:Agree. Thank you for your suggestion regarding the figures. Following your advice, we have revised the title and corresponding description of Figure 9 (now Figure 8 on Page 14) to explicitly state that the chart presents a comparison of AP values for each defect category in the liquid reservoir weld defect dataset. This clarification ensures a clear and unambiguous connection between the visual content and its corresponding dataset.

Comments 4:As of figure 10, isn't YOLOv8s with RepGFPN sufficient? When comparing images (c) and (d) in Figure 10, the results obtained by (c) is sufficiently accurate.  Also, by incorporating RepGFPN-FocalNets-CGA into YOLOv8s, you are increasing the system complexity. Please comment on this.

Response 4:Thank you for your insightful comments. As illustrated in Figure 10 (now Figure 9 on Page 16)(c), while the YOLOv8s model with RepGFPN can initially localize the defect, its attention coverage and response intensity are discernible yet comparatively weaker than those of our full model presented in (d). The heatmaps generated by our full model exhibit a more concentrated and coherent attention distribution, demonstrating its enhanced capability to accurately capture subtle defect features and their structural context.The relevant content has been added to lines 448-458 on page 17.

Although the introduction of RepGFPN, FocalNets, and the CGA module slightly increases model complexity, the ablation studies in lines 388-394 on Page 12 show that these improvements lead to only a marginal growth in parameters while significantly enhancing key metrics such as mAP@0.5 and mAP@0.5:0.95. This demonstrates that the introduced modules effectively improve the model's capability to recognize diverse defects and its generalization performance, resulting in better robustness in practical applications. Therefore, we believe the performance gains achieved within a controllable complexity range hold clear engineering value.

Comments 5:Improve the conclusion by adding statements on performance imporvement achieved. It is advised to add numerical values.

Response 5:Based on your suggestion, we have supplemented the conclusion section on Page 18 with specific performance improvement figures to more clearly demonstrate the enhancements:
Ablation studies and comparative experimental results show that the improved model achieves a 6.3% increase in mAP@0.5 and a 4.3% increase in mAP@0.5:0.95 compared to the baseline model. For key defect types, the AP values for arc craters, porosity, undercuts, and lack of fusion are improved by 3.9%, 13.5%, 5.0%, and 2.5%, respectively, effectively validating the superiority of the proposed improvements. We appreciate your pointing out the lack of specificity in our original conclusion, and have now included these quantitative results to intuitively reflect the model's performance gains.

Comments 6:Line 274 should be "Fig 7(a)" not Fig 6 (a).

Response 6:We sincerely thank the reviewer for their careful reading. The citation in question was indeed a clerical error, and we have corrected it to "Fig. 7(a)" on Page 9. Additionally, we have thoroughly reviewed all figure and table references throughout the manuscript to ensure that such errors do not occur elsewhere.

Reviewer 3 Report

Comments and Suggestions for Authors

The paper presents a well-structured and technically sound study on weld defect detection in automotive liquid reservoirs using an improved YOLOv8s model.
The methodological innovations — integration of RepGFPN, FocalNets, and Cascaded Group Attention — are well motivated and experimentally validated.
The comparative and ablation studies are thorough, showing clear quantitative and visual improvements over the baseline and state-of-the-art alternatives.

Strengths

  • Solid theoretical grounding with clear justification for each architectural modification.

  • Comprehensive evaluation across two datasets (liquid reservoir and steel pipe welds).

  • Clear visual analyses (Grad-CAM, detection maps) improving interpretability.

  • Excellent organization, figures, and reproducibility.

Suggestions for improvement

  1. Add a more explicit discussion of computational trade-offs (FPS vs. mAP) and potential strategies for lightweight deployment in industrial real-time systems.

  2. The dataset availability statement could be improved by providing at least partial open access or an anonymized subset for reproducibility.

  3. Discuss briefly the potential generalization to other welding materials or geometries.

  4. Include a brief note on limitations and possible failure cases to make the discussion more balanced.

  5. Minor textual polishing could further enhance readability, particularly in long technical sections (e.g., equations (1)–(6)).

Overall, the paper meets high academic standards and represents a meaningful contribution to intelligent defect detection in industrial inspection systems.

Author Response

Comments 1:Add a more explicit discussion of computational trade-offs (FPS vs. mAP) and potential strategies for lightweight deployment in industrial real-time systems.

Response 1:Thank you for your valuable suggestion. Following your advice, we have included a discussion of the computational efficiency vs. detection accuracy trade-off in lines 388-394 on Page 13 .

Comments 2:The dataset availability statement could be improved by providing at least partial open access or an anonymized subset for reproducibility.

Response 2: We have incorporated the Data Availability Statement in lines 527-528 on page 19 as recommended. The specific wording added is as follows: Data Availability Statement: The datasets used in this study are not publicly available. Interested researchers may obtain access by contacting the corresponding author upon reasonable request.

Comments 3:Discuss briefly the potential generalization to other welding materials or geometries.

Response 3:Thank you for your insightful comments. To evaluate the generalization capability of our proposed method, we conducted additional experiments on a public steel pipe weld defect dataset. This validation is motivated by two key factors: first, the similarity in typical defect types between steel pipes and liquid storage tank welds provides a common basis for feature learning; second, the consistent use of data from a laser profilometer ensures that the model learns geometry-agnostic defect features. The results demonstrate that our improved model maintains excellent performance on this dataset, confirming its promising cross-component applicability. These details have been added to Section 3.4.7 on page 18.

Comments 4:Include a brief note on limitations and possible failure cases to make the discussion more balanced.

Response 3: We acknowledge several limitations of our approach despite its strong performance. The relatively low AP values achieved by the improved model in detecting crater and porosity defects indicate persistent limitations. Crater defects exhibit high diversity in shape, size, and depth, while their low contrast and irregular morphology undermine feature extraction effectiveness, making it challenging for the improved model to fully capture their polymorphic characteristics. Although porosity defects possess relatively distinct circular or elliptical structures that facilitate spatial pattern recognition by the enhanced RepGFPN architecture, both the quantity and diversity of training samples for this defect category are substantially limited compared to other classes. This data distribution imbalance restricts sufficient representation learning, thereby hindering further breakthrough in detection performance. We appreciate the reviewer's suggestion and have added a discussion of these limitations in lines 411-420 on page 13.

Comments 5:Minor textual polishing could further enhance readability, particularly in long technical sections (e.g., equations (1)–(6)).

Response 5:Agree. Thank you for this valuable feedback. In response, we have thoroughly polished the language throughout the technical sections of the manuscript, particularly in paragraphs containing Equations (1) to (6), with key revisions highlighted for your convenience. While fully preserving the technical rigor, we have further refined sentence structures and technical terminology to significantly improve clarity, precision, and readability, resulting in more fluent and natural presentation.

Round 2

Reviewer 2 Report

Comments and Suggestions for Authors

Authors have significantly improve the manuscript. I am happy with the response to the reviewer comments. They are detail and also justify the authors selections